# Chitosan Nanoparticles-Based Cancer Drug Delivery: Application and Challenges

**DOI:** 10.3390/md21040211

**Published:** 2023-03-28

**Authors:** Bhuvi Sachdeva, Punya Sachdeva, Arvind Negi, Shampa Ghosh, Sungsoo Han, Saikat Dewanjee, Saurabh Kumar Jha, Rakesh Bhaskar, Jitendra Kumar Sinha, Ana Cláudia Paiva-Santos, Niraj Kumar Jha, Kavindra Kumar Kesari

**Affiliations:** 1Department of Physics and Astrophysics, Bhagini Nivedita College, University of Delhi, Delhi 110072, India; 2GloNeuro, Sector 107, Vishwakarma Road, Noida 201301, India; 3Department of Bioproducts and Biosystems, School of Chemical Engineering, Aalto University, 00076 Espoo, Finland; 4ICMR—National Institute of Nutrition, Tarnaka, Hyderabad 500007, India; 5School of Chemical Engineering, Yeungnam University, Gyeonsang 38541, Republic of Korea; 6Research Institute of Cell Culture, Yeungnam University, 280 Daehak-Ro, Gyeongsan 38541, Republic of Korea; 7Advanced Pharmacognosy Research Laboratory, Department of Pharmaceutical Technology, Jadavpur University, Kolkata 700032, India; 8Department of Biotechnology, School of Engineering and Technology, Sharda University, Greater Noida 201310, India; 9Department of Biotechnology Engineering & Food Technology, Chandigarh University, Mohali 140413, India; 10Department of Pharmaceutical Technology, Faculty of Pharmacy, University of Coimbra, 3000-548 Coimbra, Portugal; 11REQUIMTE/LAQV, Group of Pharmaceutical Technology, Faculty of Pharmacy, University of Coimbra, 3000-548 Coimbra, Portugal; 12Department of Biotechnology, School of Applied & Life Sciences (SALS), Uttaranchal University, Dehradun 248007, India; 13School of Bioengineering & Biosciences, Lovely Professional University, Phagwara 144411, India; 14Department of Applied Physics, School of Science, Aalto University, 00076 Espoo, Finland

**Keywords:** nanoparticles, chitosan, chitin, polysaccharides, nanocarriers, anticancer agents

## Abstract

Chitin is the second most abundant biopolymer consisting of *N*-acetylglucosamine units and is primarily derived from the shells of marine crustaceans and the cell walls of organisms (such as bacteria, fungi, and algae). Being a biopolymer, its materialistic properties, such as biodegradability, and biocompatibility, make it a suitable choice for biomedical applications. Similarly, its deacetylated derivative, chitosan, exhibits similar biocompatibility and biodegradability properties, making it a suitable support material for biomedical applications. Furthermore, it has intrinsic material properties such as antioxidant, antibacterial, and antitumor. Population studies have projected nearly 12 million cancer patients across the globe, where most will be suffering from solid tumors. One of the shortcomings of potent anticancer drugs is finding a suitable cellular delivery material or system. Therefore, identifying new drug carriers to achieve effective anticancer therapy is becoming essential. This paper focuses on the strategies implemented using chitin and chitosan biopolymers in drug delivery for cancer treatment.

## 1. Introduction

Chitin is the second most biopolymer composed of *N*-acetylglucosamine units. It is commonly found in higher quantities in arthropods’ exoskeletons, radula of mollusks, and cell walls of fungi [1]. Commercially, it is marketed as one of the components of natural medicinal products, nutraceutical foods, and 3D scaffolds for biomedical and technological applications [2,3]. It is typically produced using a high-temperature method and is reported to exhibit thermostability [4]. Additionally, because chitin shows high tolerance for high chemical concentrations, some metals, such as copper, can be deposited through an electrochemical process at room temperature [5]. Different forms of chitin are present in nature; the side chain’s backbone arrangement determines the difference between the forms of chitins. The α-chitin, β-chitin, and γ-chitin are the three isoforms of chitin [6].

The parallel chain positioning is found in the α-chitin structure and is usually used for tissue engineering. In contrast, an antiparallel chain positioning is found in the β-chitin structure and is used for wound healing [7]. In comparison, γ-chitin is derived from the cocoon of *Orgyia dubia* (Moth), shares structural similarities with α-chitin, and is composed of microfibers. The external morphology of *α*-chitin and β-chitin consists of nanofibers [7]. Moreover, the optimization process is responsible for chitin’s multifunctionality and structural diversity. The internal structure of chitin is an arrangement of building blocks into higher-order fiber structures, which are stabilized by the no-covalent interactions [8].

The chitin arrangement can also be seen as a crystalline structure. Chitin crystallites contain fibrillar rods generated by the supramolecular assembly comprising 19 molecular chains. The chitin rods and protein matrix form a natural composite along with the mineral content. There is considerable evidence that chitin molecules create covalent interactions with their protein matrix, forming in their original form, which includes the involvement of acetyl, amino, and hydroxyl groups in the polymer chain. Additionally, the chitin proteoglycans exhibit higher H-bond networking (intermolecular and intramolecular), resulting in densely linked chitin-derived structures. Because of such high bonding involvement, chitin has poor solubility indexing with most solvents, including water, organic solvents, and to an extent, slightly acidic or basic solutions [8]. Therefore, it serves as a robust support material for various biomedical applications. For example, chitin is used along with carbon nanotubes for neural treatment and growth [9]. It has been seen that by preventing chondrocytes from undergoing apoptosis and stimulating immunomodulation of chondrogenesis and macrophage, chitin hydrogel restored damaged cartilage [10]. Additionally, human bone marrow-derived stromal cells were utilized as a 3D-scaffold for chitin generated from demosponge *Aplysina aerophoba*, which increased the proliferation of the cell, cell bridging generation, and metabolic functions without producing much toxicity [11]. It is interesting to note that chitin is used as a possible treatment and a biological marker for neurological disorders. Chitin is increased and builds up in the brain of patients with Alzheimer’s disease (AD), providing scaffolding for the deposition of amyloid-β [12,13,14]. Brain tissues from Alzheimer’s patients also contained fungal chitin [15]. AD is one of the most recurrent neurodegenerative disorders that affect a person’s language acquisition, perception, memory, and thoughts [16,17,18,19,20,21,22,23,24,25], which leads to overall impaired cognition [26,27]. Additionally, those with multiple sclerosis had chitin deposition. A condition referred to as multiple sclerosis develops when the body’s immune system attacks the protective covering surrounding the brain’s nerve cells [28].

Chitosan (CS), a polysaccharide, is derived from chitin by deacetylation [29,30,31]. Chitosan has been reported for use in various applications, extending from biomaterials and tissue engineering to antibacterial, antifungal, anticancer, and antioxidant agents due to its strong biocompatibility [32]. Chitosan has undergone several chemical changes that have been suggested to give polysaccharides particular qualities. Chitosan samples that have been altered through phosphorylation, quaternarization, carboxylation, sulfonation, *N*-alkylation, and acylation can function as stimuli-sensitive materials (pH-, thermo-, or light-sensitive) [33]. Chitin and chitosan have been the focus of numerous investigations to determine their efficacy as agents for drug delivery [34]. For instance, chitosan is commonly used for preparing hydrogels for drug delivery due to its essential characteristics, such as bio-adhesion, having a polycationic surface that makes it easier to form hydrogenic and ionic bonds, and biocompatibility, which means it does not generate any toxins or trigger an immune response when in contact with the body fluids or living tissue [35]. Likewise, several types of research have successfully implemented chitin as one of the support materials for drug delivery. However, many ways are reported to transport the drugs, but the implementation of polymeric carriers received a high interest since they increase the effectiveness of drug targeting and extend the time that drugs stay in circulation by reducing urine elimination [36]. Herein we have emphasized the innovations and advancements in understanding the possible function of chitin and chitosan biopolymers in delivering cancer-treating drugs.

## 2. Extraction of Chitin: Chemical and Biological Process

The cuticles of different crustaceans, primarily shrimp and crabs, are the prime producers of raw material for the formation of chitin. Chitin is a complex protein network component mainly found in crustaceans or shellfish, on which the calcium carbonate accumulates to produce a firm shell [37]. Chitin and protein interact very closely; a minor amount of protein is also contained in a polysaccharide-protein complex. The extraction of chitin from shelf fish primarily requires two-step chemical processing (a) removal of inorganic calcium carbonate by demineralization and (b) removal of proteins by deproteinization. Sometimes, a different decolorization phase is also used to eliminate any remaining pigments [38]. Heat and high alkaline or acidic conditions (>1 M NaOH, >3 M HCl) can cause chitin to change itself into the deacetylated state, chitosan, or hydrolyze into C5 and C6 hydrocarbons [39] on prolonged chemical treatments [1]. Figure 1 shows chitin extraction and chitosan production chemical and biological processes. However, several methods of producing pure chitin have been devised, but no method is accepted as standard procedure to this date. Deproteinization and demineralization could also be accomplished by enzymatic (biological) or chemical processes. Additionally, microbial fermentation is used to simultaneously carry out the demineralization and deproteinization processes [40]. Biological processes, an alternative approach, may produce more satisfactory results because they are inexpensive and environmentally friendly, have low energy usage, and are reproducible. In addition, they can extract or manufacture chitin with greater molecular weight and a more robust crystal structure [41].

### 2.1. Chemical Process

#### 2.1.1. Chemical Demineralization

Demineralization is a process of mineral removal, typically calcium carbonate. Demineralization is usually accomplished through acid treatment with H_2_SO_4_, HNO_3_, CH_3_COOH, HCOOH, and HCl. Dilute hydrochloric acid is the preferred reagent among these acids [42]. Demineralization is simple as it involves breaking down calcium carbonate into calcium salts (water-soluble) with carbon dioxide emission [43]. Treatments involving demineralization are usually empirical and rely on the specificity of samples, including the degree of shell’s mineralization, particle size, temperature, extraction time, solute or solvent ratio, and acid content [44,45].

#### 2.1.2. Chemical Deproteinization

The chemical interactions between chitin and proteins must be broken during the deproteinization process, which is challenging. The biopolymer is depolymerized during this heterogeneous process employing chemicals. For biomedical applications, the total elimination of protein is crucial since a portion of the population suffers from shellfish allergies, with the protein component being the only problem. The first strategy for deproteinization was using chemical reagents. Chemicals, such as K_2_CO_3_, KOH, Ca(OH)_2_, NaOH, Na_2_CO_3_, NaHCO_3_, NaHSO_3_, Ca(HSO_3_)_2_, Na_3_PO_4_, and Na_2_S, have been examined as deproteinization reagents. Each study’s reaction conditions are very different. The preferred reagent, NaOH, is used with different concentrations (0.125 M to 5.0 M), temperatures (up to 160 °C), and treatment times (from a few minutes up to a few days). The NaOH treatment causes deproteinization, and partial deacetylation of chitin, resulting in a decrease in molecular weight [46,47].

#### 2.1.3. Depigmentation

In arachnoids, there are mineral-free cuticles; therefore, the process can start without the demineralization step, and isolation of chitin can begin with deproteinization and depigmentation [48]. Firstly, the lipids and waxes must be removed from the cuticles, which is achieved through microwave irradiation (MWI) [48]. In MWI, cuticles are introduced to microwave radiation while they are treated with ethanol and chloroform. In the second step, microwave radiation is used with NaOH to remove the pigments and other proteins from the cuticle [48]. Figure 1 shows the chemical process of chitin extraction.

### 2.2. Biological Process

The application of proteolytic enzymes for the digestion of proteins or the fermentation process allows the digestion of proteins and minerals are the two most typical biological methods for the extraction of chitin. This method includes enzymatic deproteinization in which proteases (pepsin, papain, alcalase, trypsin, etc.) are used to remove protein [49]. This process of deproteinization can be performed either before or after demineralization. The method of deproteinization can be conducted through the process of fermentation as well. Further, the depigmentation process takes place to extract pure chitin [46]. Figure 1 shows the biological process of chitin extraction.

## 3. Drug Delivery System

Combining an appropriate carrier with one or more drugs is the basic building block of drug delivery systems [50]. The two main criteria for perfect delivery systems are concentrating the active substance on the body’s site of action and supplying the correct quantity for a steady, suitable, and predetermined amount of time [51]. The first fundamental element of drug delivery systems is keeping plasma drug concentrations under the therapeutic window, accomplished through drug carriers, typically polymers [51]. This feature helped to promote the notion of controlled delivery systems. The second characteristic results in the creation of specialized delivery systems. The active substance is captured within a delivery system that may deliver the medicine to the particular site, necessitating careful selection of carrier, route of administration, and target of release [52]. The qualities of the medication and polymers, the disease to be treated, the variety of dosage forms, and the method of administration are just among the variables that affect whether delivery systems succeed in achieving their therapeutic goal [51]. As carriers, active compounds are frequently delivered using polymers (such as chitin or chitosan). They can be utilized to create various delivery systems, including micelles, hydrogels, tablets, capsules, and particulate systems (beads, micro-, and nanoparticles). The optimized pharmacokinetics usually get considered before choosing an appropriate polymer as it dictates the kinetics of drug release and the removal of carriers following drug delivery [53]. In a study by Machalowski et al. [42], chitin was utilized to deliver the aceclofenac drug through the topic and transdermal routes. Similarly, Sabitha et al. [54] and Panonnummal et al. [55] utilized chitin as nanogels to deliver 5-fluorouracil and clobetasol through the topical and transdermal route. Chitosan can take numerous ways the delivery of drugs. The routes of chitosan administration for drug delivery have been elaborated in detail further in the below sections.

### 3.1. Rotes of Chitosan Administration

#### 3.1.1. Ocular Drug Delivery of CS

As mentioned above, chitosan’s biodegradability and biocompatibility are important features, making it useful for ocular drug delivery. Chitosan can increase the ocular surface duration of several drugs due to its mucoadhesive nature [56]. It can also transform to gel if smeared on the ocular surface in liquid form as it possesses favorable situ gelling properties. This has led to the therapeutic improvement of ocular drugs. The ocular drugs which are poorly soluble, chitosan nanoparticles (NPs) are a potential alternative, as they can increase the bioavailability of drugs (e.g., naringenin) in the aqueous humor. Moreover, a study by Ping Zhang et al. on rabbit eyes found that chitosan has no irritating effect on the eye [57]. Fluconazole-loaded chitosan NPs were created by Santhi et al., utilizing a cross-linking approach and spontaneous emulsification. They compared the antifungal abilities of these NPs with the traditional eye drops using the cup-plate method. These particles had an average size of 152.85 ± 13.7 nm. All drug-loaded NPs were determined to have an optimal (50%) drug-loading capacity. After completing their research, they deduced that the fluconazole-formulated chitosan NPs were an effective delivery system for fluconazole in drug loading, antifungal efficacy, and prolonged release characteristics [58].

#### 3.1.2. Pulmonary Drug Delivery of CS

The benefits of delivery of drugs to the lungs include immediate and prolonged drug delivery, high effectiveness, and the ability to accomplish both local and systemic effects. Large lung surface area, high vascularity, and a thin absorption barrier are the parameters that improve medication transport via the lungs [59]. Chitosan has been used to enhance the effects of many medications. Rifampicin, an antitubercular medicine, was created as a dry nanoparticle powder inhalation using chitosan as the polymer. This formation demonstrated continuous drug release for up to twenty-four hours without causing any adverse effects on cells or organs [60]. Prothionamide, an antitubercular medication, was given by Debnath et al. as chitosan-coated NPs through the lungs. This change lengthened the drug’s inhalation half-life in the lungs [61]. Itraconazole is an antifungal drug, and due to itraconazole’s poor oral solubility, Jafarinejad et al. produced chitosan NPs for the pulmonary delivery of the antifungal medication as a dry powder formulation. They increased the drug’s aerosolization capabilities by adding chitosan NPs, mannitol, lactose, and leucine to the formulation. As a result, there was an increase in itraconazole pulmonary deposition [62].

#### 3.1.3. Mucosal Drug Delivery of CS

Chitosan and its derivatives encourage mucosal delivery by increasing the absorption of hydrophilic molecules such as protein and peptide medicines. The eminently hydrated glycoproteins (lysozymes, salts, and mucins) that make mucus give it its viscoelastic characteristics [63]. To facilitate the paracellular trafficking of macromolecular medicines, chitosan function by opening the compact intercellular junctions. The positively charged, cell-bound chitosan NPs reduce the transepithelial electrical resistance of living cell monolayers and boost paracellular permeability. Depending on the chitosan’s molecular weight and level of deacetylation, the chitosan solution enhances trans and paracellular permeability [64]. Chitosan’s positive charge interacts with the compact junction proteins ZO-1 and occludin, redistributes F-actin, and somewhat destabilizes the plasma membrane as part of its mechanism of action. The environment has also been demonstrated to affect chitosan’s capacity to increase penetration [65].

#### 3.1.4. Nasal Drug Delivery of CS

Using a non-invasive method like nasal delivery, medications can be administered systemically and locally without experiencing the normal gastrointestinal problems associated with oral management or the effects of hepatic metabolism [66]. Although nasal management can cross the blood–brain barrier (BBB), which has been shown to guide drug delivery from the nose to the brain (NTB) efficiently, NTB is an alternate way of topical management for antibacterial and anti-inflammatory nasal congestion [29]. One of three pathways (three methods of nasal absorption) allows nasally administered drugs to instantly cross the BBB: first, olfactory nerves, which are the foremost effective pathway for NTB delivery of drugs; second, trigeminal nerves, which have the presence of nerve endings in the respiratory epithelia; and third, respiratory epithelium. Some limitations of NTB delivery include the minor volume of the nasal cavity, enzymatic degradation, mucociliary clearance, short drug retention duration, potential nasomucosal toxicity, drug management and deposition technique, and the need for an appropriate delivery device [67]. Due to their limited permeability, the nasal epithelium is challenging to penetrate with hydrophilic medicines, nucleic acids, proteins, and peptides. CS enhances their permeability. The drug’s weight, lipophilicity, and charge affect how well it is absorbed through the nose. The mucociliary system clears medications that cannot pass the nasal membrane. Due to its use in nasal delivery, CS has mucoadhesion qualities combined with low toxicity, biodegradability, and biocompatibility, which can assist in resolving this concern [66].

Moreover, high molecular weight molecules cannot pass through the BBB and blood-cerebrospinal fluid barrier (BCB) into the brain. However, NTB delivery has been a vital strategy in the last few years to overcome these challenges and deliver drugs to the targeted brain regions. Human serum albumin NPs (HSA NPs) coated with chitosan were created by Piazzini et al. functioning as a nose-to-brain carrier for the anti-Alzheimer medicines tacrine and R-flurbiprofen. On ex vivo rabbit nasal mucosa, CS-coated NPs demonstrated improved mucoadhesion and a higher penetrating capacity than uncoated NPs. Additionally, by lowering the levels of ZO-1 expression, they have the benefit of loosening the compact junctions between hCMEC/D3 cells, allowing molecules to pass across the barrier [67].

To increase drug concentration in the active site, direct therapeutic material delivery to the brain is necessary for neurologic illnesses such as Parkinson’s disease (PD). In the central nervous system, PD is characterized by neurodegeneration and dopaminergic neuron loss in the CNS. The current standard of care for managing PD motor symptoms relies on the dopamine (DOPA) replacement strategy, which tries to compensate for the death of dopaminergic neurons and restore adequate neurotransmitter levels. Due to elevated hydrogen-bonding potential, complete ionization in physiological pH, and significant metabolism when administered orally, it is challenging for DOPA to penetrate the BBB. The development of DOPA-loaded nanocarriers as a novel mechanism for treating Parkinson’s disease has received the most significant attention [68]. These nanocarriers should have the capability to traverse BBB and permit persistent transport of the neurotransmitters to the brain. A non-ergoline agonist which works on the brain’s D2- and D3 receptors is ropinirole hydrochloride (RH). Chatzitaki et al. created PLGA (poly(lactic-co-glycolic acid) and PLGA/CS NPs with mucoadhesive characteristics to increase RH transport through the nasal mucosa. RH-loaded PLGA/CS NPs demonstrated full drug release over 24 h in the simulated nasal electrolyte solution (SNES) [68].

#### 3.1.5. Transdermal Drug Delivery of CS

A transdermal drug delivery system is being evolved to overcome the shortcomings of traditional administration methods. The limited skin permeability is the fundamental obstacle to be addressed when creating transdermal dosage forms. Several techniques have been devised to get around the barrier qualities and improve the transportation of medication molecules over the skin [69,70,71]. Numerous transdermal patches made of polysaccharides have been discovered in recent years. Transdermal preparations containing CS are becoming more widespread [71]. NPs have been promoted as one of the prospective delivery systems that can significantly overcome the constraint of the drug’s ability to penetrate the skin. As a transdermal drug delivery technique, polysaccharide NPs are becoming increasingly well-liked. The mucoadhesive, biocompatible, mucodegradable, and mucosal penetration enhancing properties of the CS NPs. They are interrelated with the skin mucosa and fluidize the lipid and protein domains of the epidermis to aid transdermal drug diffusion. Additionally, they might be beneficial therapeutically for local illnesses such as skin infections and malignant melanoma and systemic conditions such as hyperlipidemia and diabetes [72].

#### 3.1.6. Dermal Delivery of CS

The systemic unpropitious effects of traditional oral and injectable delivery could be avoided with topical treatment. Additionally, this can swiftly and directly penetrate the skin and mucous membranes at the illness site [29]. Since they enable the regulated release of drugs and address the issue of their low skin bioavailability, NPs are seen favorably for treating acne. Nicotinamide is one of the potential cosmeceuticals/nutraceuticals lately utilized to treat acne. This medication has anti-inflammatory effects and is said to reduce sebum production. Abd-Allah et al. produced CS NPs as well as added nicotinamide-containing supplements. On individuals with acne vulgaris, the NPs were refined, described, and tested in a clinical setting. Cogent skin adherence ex vivo and elevated nicotinamide deposition in the various skin layers totaling 68%, were used to demonstrate the topical benefits of CS NPs [73]. Furthermore, the nicotinamide CS NPs showed 73% depletion in inflammatory acne lesions when clinically evaluated on patients in contrast to untreated areas, demonstrating that the delivery system could be a therapeutically viable alternative for treating skin diseases [73].

#### 3.1.7. CS Administration for Wound Healing

Various bacteria can infect and colonize injured skin, making it easier for them to get to the underlying tissues [74]. One crucial element that is thought to slow the healing of wounds is infection. In addition to providing a moist surrounding to prevent wound dryness, reducing wound surface necrosis, being oxygen penetrable without dehydrating the wound, and being congenial, wound dressings should also prevent mechanical damage [75]. Less toxicity, biocompatibility, and biodegradability are further important requirements for a material used to make wound dressing [76]. It has been demonstrated that the N-acetyl glucosamine, which is a monomer unit of CS, promotes cell growth, promotes hemostasis, as well as speeds up the healing of wounds.

Regarding biocompatibility, CS has no adverse effects on touch with human cells [77]. Additionally, CS speeds up blood coagulation by attaching to red blood cells. Finally, the constancy of the medications by creating CS NPs increases drug aggregation [77].

## 4. Cancer: Symptoms, Causes, Treatment Strategies

Cancer is a disease in which a group of the cells of the body starts growing and spreading abnormally [78]. The normal and healthy human cells multiply and grow through a process known as cell division, which allows the formation of new cells for the body. When cells are damaged or become old, they die and are replaced by new cells in the body. However, sometimes this process disrupts, leading to the growth and proliferation of abnormal cells (cancerous cells) [79]. In cancer, continuous clonal expansion of cells (somatic cells) is destroyed by destabilizing, eroding, and invading the healthy tissues [80]. The involvement of prior diagnosis assessments in worldwide cancer regulation programs, including symptom recognition campaigns, is expanding. However, if the symptoms specify the early stage of the disease, these strategies will have a nominal effect on refining cancer outcomes [81]. In a study by Koo et al., 7997 patients having cancer (stage IV cancer) were analyzed to examine the symptoms. Some typical symptoms were post-menopausal bleeding, breast lump, abnormal mole, fatigue, weight loss, abdominal pain, rectal bleeding, hoarseness, bowel habit change, haematuria, and symptoms associated with the lower urinary tract [81]. Some of the common causes related to the development of cancer are shown in (Figure 2). There are several strategies for the treatment and early detection of cancer. Some of these are listed in Table 1.

## 5. Chitin and Chitosan for Drug Delivery and Cancer Treatment

Drug development and delivery have seen significant breakthroughs due to nanotechnology. For instance, the utility of NPs in the treatment and diagnosis of cancer has advanced to the point where it can now detect and target a single cancer cell with the delivery of a carrier to treat it. Traditional cancer therapeutic techniques include side effects, and diagnostic procedures are expensive and time-consuming. Due to their large size, surface charge, and morphology, NPs such as carbon nanotubes (CNTs), calcium NPs (CaNPs), graphene, and polymeric NPs (including chitin and chitosan) have improved cancer diagnostics and treatments. These NPs functionalization with various biological molecules, such as antibodies, aids in the transportation of drugs and the detection of cancerous cells [88]. Chitin holds the ability to generate as a drug delivery system and anticancer agent. It has been demonstrated that chitin can suppress chitinase-3-like protein-1 (CHI3L1), which is overexpressed and stimulates proinflammatory mediators in breast cancer cells [89]. Moreover, the synthesis of vascular endothelial growth factor C (VEGF-C), associated with tumor angiogenesis, can be downregulated with chitin [90]. Chitin has been formed in several kinds that can counteract cancer. For example, cytotoxicity was promoted in human breast cancer cells (MCF-7 Cells) with chitin nanocomposites embedded with silver [91]. Curcumin is an active turmeric substance with anticancer, antibacterial, and antifungal properties [92]. Curcumin-loaded chitin nanogels (CCNGs) is an anticancer drug with chitin and curcumin and are insoluble in water. It has been seen that CCNGs-prepared materals, when treated on porcine skin samples, showed easy penetration in the epidermis od the skin with no signs of inflammation. This shows that the formulation of CCNGs can treat melanoma, which is one of the most serious and common types of skin cancer [93]. Cancer vaccine has evolved as a unique cancer treatment method with the emergence of cancer immunotherapy, and the significance of adjuvants has lately been recognized. Adjuvants are chemical compounds that boost immunity and promote a vaccine’s potency without exhibiting any direct antigenic consequences of their own [94]. In addition to the previously listed applications, chitin and chitosan are essential adjuvants for immunotherapy. Many studies have investigated the adjuvant characteristics of chitin and chitosan due to their immunostimulant capability and structural resemblances to glucans, a subsidiary type of natural polysaccharides [95]. Chitin and chitosan’s antiviral and anticancer properties were first described decades ago. Suzuki et al. initially showed the adjuvant action of chitin and chitosan in the 1980s [96]. Chitin and chitosan are frequently used for non-invasive mucosal management routes, such as oral, intranasal, and ocular mucosa, due to their mucoadhesive characteristics [97]. Specific antigens have been demonstrated to boost adaptive immune responses [97]. Recent studies have shown that chitosan is a potential adjuvant for intranasal vaccination [98].

Moreover, chitin has a size-dependent and complex effect on adaptative and innate immune response, including the capability to activate and recruit innate immune cells, which stimulates chemokine and cytokine production [99]. It has been seen that IL-12 is an antitumor cytokine that induces toxicity upon systematic administration. IL-12 can be formulated with chitosan (chitosan/IL-12) and administrated (intratumorally) in tumor mice model, could help in limiting the systematic toxicity by enhancing the local retention in the tumor microenvironment of the IL-12 [100]. Some of the chitin and chitosan-based nanocomposites for drug delivery and treatment of various types of cancer have been mentioned in (Table 2). Moreover, the delivery of most of the NPs or nanocarriers, including chitin and chitosan, is of two types: the passive targeting of NPs for drug delivery and the active targeting by NPs for drug delivery. The passive drug delivery of chitosan for cancer treatment is explained in (Figure 3) and the active drug delivery of chitosan for the treatment of cancer is explained in (Figure 4).

One of the critical findings of passive targeting is the leaky vasculature or lymphatic drainage of the tumors. This finding has led to a significant conception of EPR (enhanced permeability retention) effect. A neovasculature system is formed by metabolic cancer cells in which the blood vessels are irregular and leaky (exhibit inter-endothelial gaps) compared to healthy vessels. The inter-endothelial gaps help transport nanocarriers and NPs through the formation of paracellular pathways. The tumor cell endothelium has a concentration gradient that further facilitates the accumulation of NPs or, in this case, anticancer agents released by CS nanocarriers into the tumor. Once the anticancer agents are taken by cancer cells, the cell will go under apoptosis. The concentration, size of NPs (less than the size of inter-endothelial gaps of vessels), and blood circulation are the most crucial features responsible for passive targeting.

## 6. Advantages of Using Chitin and Chitosan in Nanomedicine

### 6.1. Biocompatibility

Chitin and chitosan can be utilized effectively in the human body without having any negative effects because they are biodegradable and biocompatible. They are, therefore, suitable for a range of biomedical applications, such as wound healing, tissue engineering and medication delivery. Chitin and chitosan have been utilized to create polymer scaffolds. Furthermore, there is growing interest in using chitosan to create nanocarriers and facilitate microencapsulation techniques for the transport of medications, biologics, and vaccines [32,130]. The chitin and chitosan are useful as they can be created as chitin or chitosan-based nano- and micro-particles with certain sizes and cargo-release properties [130].

### 6.2. Antimicrobial Characteristics

Research has revealed that chitin and chitosan exhibit antimicrobial action against a variety of pathogens, including fungi, bacteria, and viruses. Due to this characteristic, they can be used to create antimicrobial coats for medical equipment and to treat infectious conditions [131,132,133]. Moreover, chitosan films have also been employed as a packaging material to maintain the quality of a wide range of food items [134]. Chitosan exhibits strong antibacterial properties against both Gram-negative and Gram-positive bacteria, fungus, and other pathogenic and spoilage micro-organisms [135,136].

### 6.3. Mucoadhesive Characteristics

Chitosan and chitin contain mucoadhesive capabilities, which means they could cling to mucosal membranes that are found in the nose, mouth, and gastrointestinal system. Chitosan and chitin can be developed into medication delivery systems for oral and nasal administration because of this characteristic. The use of chitin and chitosan as the vehicles for mucoadhesive system drug delivery has a significant impact that further emphasizes the potential advantages of increased therapeutic agent bioavailability, prolonged drug residence time at the site of administration, and comparatively quicker drug absorption into the systemic circulation [137].

### 6.4. Biodegradability

Chitin and chitosan can be converted into non-toxic chemicals by the body’s natural mechanisms since they are biodegradable. They are the perfect choice for use in drug delivery systems that call for a sustained drug release over a long timeframe because of this attribute [9]. According to current research, lysozyme and bacterial enzymes in the colon are the main degraders of chitosan in vertebrates [138]. Many different microbes produce and/or breakdown chitin [139].

## 7. Problematics of Chitin and Chitosan in Nanomedicine

### 7.1. Allergenicity

It has been demonstrated that chitin and chitosan can cause allergic reactions in some people by inducing an immunological response. Therefore, its usage in some biological applications may be constrained by this fact. Unfortunately, chitin-chitinase-stimulated hypersensitivity is a common cause of occupational allergy. Moreover, current research has studied the immunologic effects of chitin both in vivo and in vitro, and these investigations have shown new facets of how chitin regulates innate and adaptive immune responses. It has been demonstrated that exogenous chitin controls adaptive type 2 allergic inflammation in addition to activating macrophages and other innate immune cells. These results further show that chitin interacts with many cell surface receptors, including the macrophage mannose receptor, to activate macrophages [140].

### 7.2. Limited Solubility

Chitin and chitosan’s utility in various applications may be restricted by their inability to dissolve in neutral pH water. Yet, by altering their chemical makeup or using the right solvents, solubility can be increased. The degree of acetylation, pH, temperature, and polymer crystallinity are some of the variables that affect how soluble chitosan is [141]. The lower solubility of chitosan was attributed to the polymer’s increased crystallinity following deacetylation, which counterbalances the effect of the polymer’s increased glucosamine moieties. On the other hand, the half-acetylated sample showed a decrease in crystallinity. The solubility window of chitosan is also changed by the application of hydrogen bond disruptors such as urea or guanidine hydrochloride. In actuality, wide solubility is accomplished by combining chemical and physical disruption of the hydrogen bonds [142].

### 7.3. Variability from Batch to Batch

Depending on the source and preparation techniques used to create chitin and chitosan, its characteristics can change. This may result in batch-to-batch variability, which may have an impact on some applications’ ability to reproduce and maintain consistency in their results [143].

### 7.4. Limited Stability

Recently, a lot of studies have been put into creating reliable and safe chitosan products. Unfortunately, the issue of chitosan-based systems’ weak stability limits their practical applicability; as a result, it has become extremely difficult to produce chitosan formulations’ adequate shelf-life [141,143]. The degree of chitosan purity has a significant impact on the substance’s solubility and stability in addition to its biological characteristics such as immunogenicity or biodegradability. Moreover, chitosan’s stability is affected by a number of variables, including the degree of deacetylation, moisture content, and molecular weight. Similarly, the stability of chitin is also limited; however, cross-linking chitin with enzymes or other chemical compounds can help in the upgradation of the stability of chitin [141].

## 8. Challenges and Future Perspectives

The use of chitosan-based medication carriers for cancer treatment is becoming more popular. The chitosan-based nanocarriers appear to hold promise for clinical translation. Chitosan nanoparticles are drug carriers with good biocompatibility and can be easily modified during synthesis [31]. One of the most significant obstacles to chitosan nanoparticle clinical application is its insufficient hemocompatibility. Several studies have shown that chitosan has hemostatic action. Although animal studies have indicated that chitosan-based nanoparticles are tolerable in intravenous injections, deadly emboli formation is dangerous. Off-target distribution of chemotherapeutic drug-loaded chitosan nanocarriers may also represent a clinical translational issue. The reticuloendothelial system, which circulates throughout the body, easily phagocytoses these carriers [104]. Its pharmacological cargo may be delivered to multiple organs during apoptosis. Although the FDA has approved chitosan for oral medicine delivery and wound healing, drug compositions with chemical changes of chitosan may cause in vivo toxicity [144]. Various other pharmaceutical applications have also been cited with chitosan nanoparticles which improve the physicochemical properties of active medicinal agents [39]. As a result, safety assessments for formulating chitosan-based drug carriers and evaluation of their nanotheranostic platforms are critical [145]. Despite these obstacles, chitosan has significant potential in cancer treatment. Current restrictions will be addressed when new technologies emerge and an understanding of the mechanism of action of chitosan-based drug delivery carriers evolves.

## 9. Conclusions

Cancer is a group of diseases characterized by abnormal cell proliferation and the ability to spread to other body parts, in contrast to benign tumors, which remain stationary. Possible warning signs of cancer include a lump, unusual bleeding, a persistent cough, weight loss, and a change in bowel habits. While these symptoms may indicate the presence of cancer, there could also be other causes. Over 100 different types of cancer can affect humans, and various therapeutic strategies are available for their management and treatment, including the use of nanoparticles. Chitin and chitosan are polymeric nanocarriers that can assist in the delivery of anticancer agents and inhibit the growth of cancer cells. These biopolymers have shown promise in drug delivery due to their biodegradability, biocompatibility, and low toxicity. Chitosan, in particular, has attracted attention as a potential carrier for anticancer drugs due to its ability to target cancer cells and enhance drug efficacy. However, further research is needed to explore the full potential of chitin and chitosan in cancer treatment.

In conclusion, chitin and chitosan biopolymers have emerged as promising materials for drug delivery in cancer treatment. These polysaccharides have demonstrated favorable properties, including biodegradability, biocompatibility, and low toxicity, making them ideal candidates for use as drug carriers. Moreover, chitosan, in particular, has been shown to have the ability to target cancer cells and enhance drug efficacy, indicating its potential in the development of anticancer therapeutics. However, several challenges must be overcome before these biopolymers can be successfully used in clinical settings. For instance, optimization of the drug loading and release properties of chitin and chitosan carriers is crucial to ensure their effectiveness in delivering therapeutic agents to cancer cells. Further studies are also needed to elucidate the mechanisms underlying their antitumor effects and identify any potential adverse effects. Despite these challenges, chitin and chitosan biopolymers hold significant promise in the development of more effective and safer anticancer drugs. With continued research and innovation, these biopolymers may become an essential tool in the fight against cancer.

## Figures and Tables

**Figure 1 marinedrugs-21-00211-f001:**
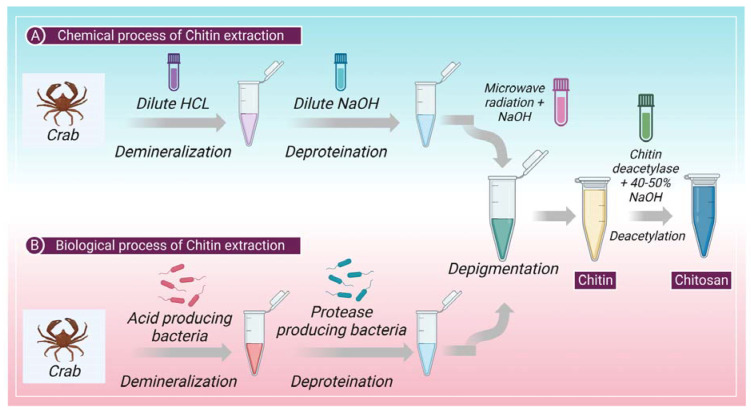
Shows the (**A**) chemical and (**B**) biological process involved in the extraction of chitin and production of chitosan from chitin.

**Figure 2 marinedrugs-21-00211-f002:**
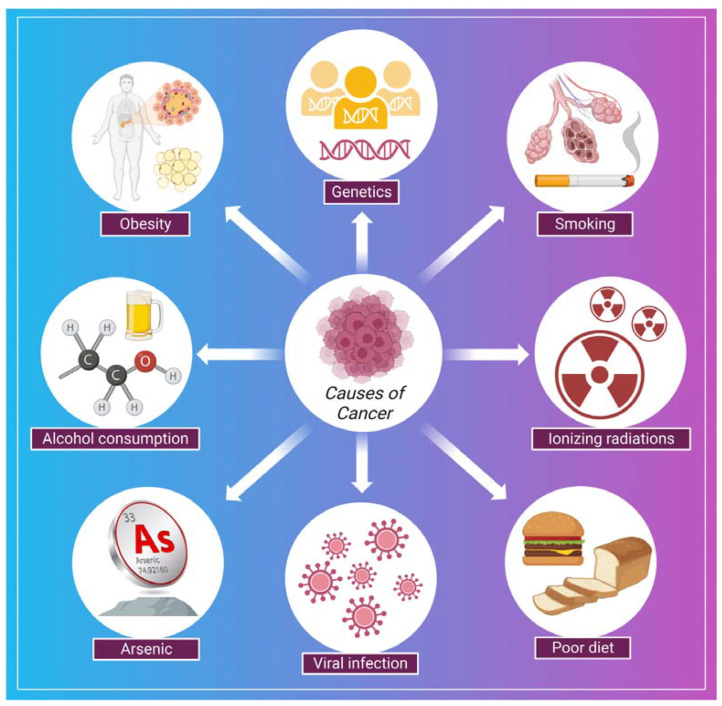
Shows some of the common causes of cancer.

**Figure 3 marinedrugs-21-00211-f003:**
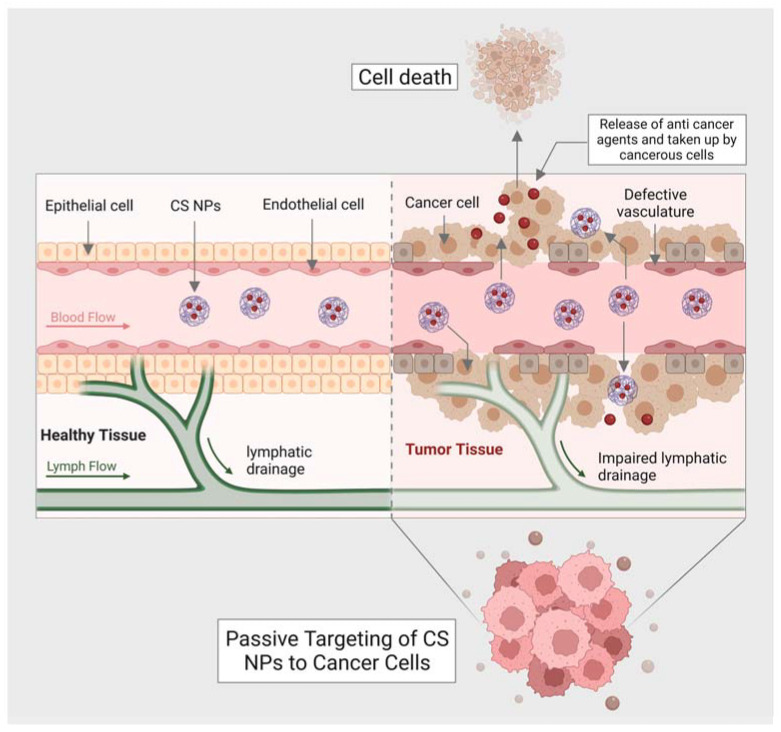
Shows passive targeting of CS nanocarriers for drug delivery against cancer cells.

**Figure 4 marinedrugs-21-00211-f004:**
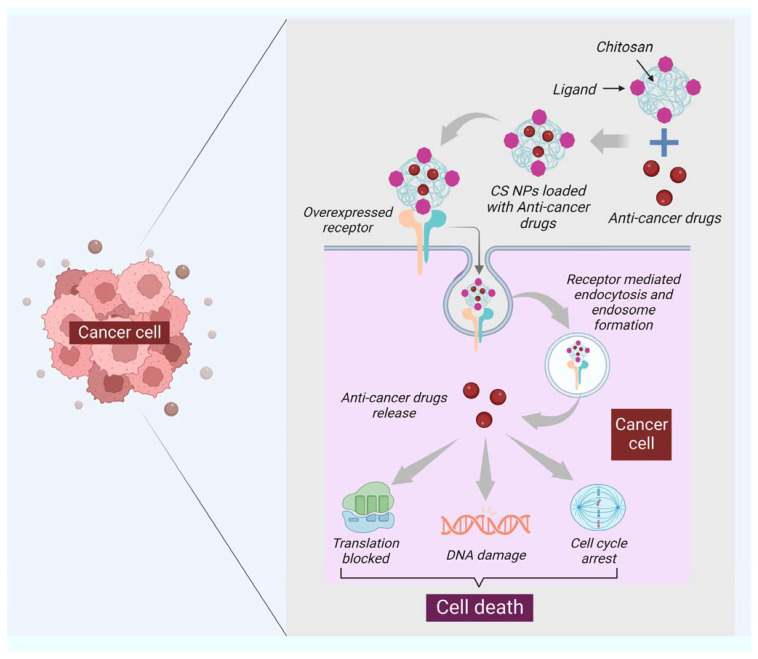
The functionalization of the surface of NPs specific to the ligand is one of the critical features of active targeting. The ligand selected should be specific to the overexpressed receptor at the surface of cancer cells. The figure shows that the ligand of chitosan nanocarriers binds to the overexpressed receptor of cancer cells; after binding, the receptor-mediated endocytosis takes place, leading to the formation of the endosome, and then the dependent release of drugs will happen. Once the pill is released, the cell will proceed under apoptosis through DNA damage, translation block, and cell cycle arrest.

**Table 1 marinedrugs-21-00211-t001:** Shows some of the standard methods used to treat cancer.

S.No	Cancer Treatment	Method	Ref.
1.	Biomarker detection	Biomarkers such as proteins, sugars, nucleic acids, cytokinetic and cytogenetic parameters, and entire tumor cells, which are sometimes found in the body’s fluid, is used for the treatment, prognosis, and diagnosis of cancer.	[82]
2.	Surgery	Surgery, the oldest oncological discipline, helps preserve function, quality, and form of life. It is a procedure in which a surgeon removes cancer from the body using invasive surgical tools.	[83]
3.	Photodynamic therapy (PDT)	In PDT treatment, a photosensitizing agent (drugs) and light kill the cancer cells. The photosensitizing agents can be administered into the bloodstream or put directly on the skin. This depends upon which body part for cancer is getting treated.	[84]
4.	Radiation therapy	In this therapy, a high dose of radiation is applied to inhibit cancer cells from its multiplication, and it can also help shrink the tumor size.	[85]
5.	Immunotherapy	The immune system concedes aberrant cells, eradicates them, and most likely ceases or decelerates the growth of numerous malignancies as part of its consistent activity. Immune cells, for example, can frequently be determined around malignancies. These lymphocytes, also called tumor-infiltrating lymphocytes, or TILs, confirm that the immune system recognizes the tumor. People dealing with cancers that have TILs are frequently better than those whose tumors do not have.	[86]
6.	Chemotherapy	Chemotherapy refers to the use of medications to kill cancer cells. This cancer medication prevents cancer cells from growing, dividing, and proliferating. Chemotherapy is a systemic treatment. This implies it circulates throughout the body via the bloodstream. Chemotherapy comes in a variety of forms. Chemotherapy medications are potent chemicals that treat cancer by destroying cells at various stages of the cell cycle. The cell cycle is the process of forming new cells in all cells. Because cancer cells develop faster than normal cells, chemotherapy has a greater impact on these rapidly expanding cells.	[87]

**Table 2 marinedrugs-21-00211-t002:** Shows results of experiments performed using chitosan and chitin biopolymers and conjugating or loading them with different types of chemical compounds. The result shows that several conjugated/loaded or encapsulated chemical compounds with chitosan or chitin can be a potential drug for treating various types of cancer and tumor.

S.No.	Chitin/Chitosan Biopolymer	Encapsulated/Loaded/Conjugated Compound	Cell Line/Animal Model	Cancer Type	Result	Ref.
1.	Chitin	AgNPs	HepG2 cells (HB-8065)	Liver cancer	The HepG2 cell line was significantly affected by the produced AgNPs. Additionally, HepG2 cells treated with AgNPs showed increased expression of apoptosis-related proteins such as Bax, cytochrome-c, caspase-3, caspase-9, and PARP and decreased expression of anti-apoptotic proteins Bcl-2 and Bcl-xL. Therefore, the results of this work indicate that biologically produced AgNPs have anticancer action against HepG2 cells and may be crucial in the future development of novel cancer therapeutics.	[101]
2.	Chitosan	Curcumin	4T1 cell line and placental-derived mesenchymal stem cells (PDMSCs)	Triple-negative breast cancer (TNBC)	According to findings, TRAIL (tumor necrosis factor-related apoptosis-inducing ligand) expressing PDMSCs and curcumin nanoparticles delivered concurrently effectively causes apoptosis in tumor cells and substantially limits tumor development in vivo.	[102]
3.	Hyaluronic acid (HA)-modified chitosan nanoparticles (CS NPs-HA)	Cyanine 3 (Cy3)-labelled siRNA (sCS NPs-HA)	A546 human cells and female BALB/c mice.	Lung cancer	The tumor growth was inhibited through the downregulation of BCL2 (*B-cell lymphoma 2)*	[103]
4.	HA-CS-NPs	Co-encapsulation of doxorubicin (DOX) with miR-34a	MDA-MB-231 cells and female BALB/c mice (athymic nude)	Breast cancer	miR-34a can inhibit the migration of breast cancer cells via targeting Notch-1 signaling.	[104]
5.	Mitomycin-C	Chitosan	T24 cell line of bladder cancer	Bladder cancer	Chitosan encapsulates in mitomycin-C showed a decline in the tumor cell activity.	[105]
6.	Low molecular weight (LMW) chitosan	2-acrylamide-2-methylpropane sulphonic acid (AMP)	A549 (lung adenocarcinoma), HepG_2_ (hepatocellular carcinoma), HeLa (Cervical Carcinoma) and Balb/c mice model	Lung, cervical and liver cancer	Increased transfection efficiency was seen in cancer cells (A549, HepG_2_, HeLa), and in the mice model, high luciferase expression was demonstrated.	[106]
7.	Quaternized chitosan-gallic acid-folic acid stabilized gold nanoparticles (Au@QCS-GA-FA)	3,4,5-tribenzyloxybenzoic acid (GAOBn)	CHAGO cells	Lung cancer	Through active targeting of cancer cells, the combination Au@QCS-GA-FA/GAOBn demonstrated remarkably effective cellular absorption and localization of gold nanoparticles. This showed the potential of Au@QCS-GA-FA as a carrier system for lung cancer treatment that targets the delivery of anticancer agents.	[107]
8.	Thiolated glycol chitosan	Pgp-targeted poly-siRNA (psi-Pgp)	MCF7/adriamycin-resistant breast cancer cell type	Human breast adenocarcinoma	Thereafter intravenous treatment, the psi-Pgp-tGC NPs accumulated in MCF-7/ADR tumors and downregulated P-gp expression to sensitize cancer cells.	[108]
9.	poly(ethylene glycol) -chitosan	small interfering RNA (siRNA)	Murine 4T1 (Mammary tumor cell line of the mouse)	Breast cancer	The siRNA-carrying PEG-chitosan nanoparticles were effectively absorbed by cancer cells, leading to anticancer activity in xenografts.	[109]
10.	biotinylated chitosan-graft-polyethyleneimine (Bio-Chi-g-PEI)	siRNA	Hela and human ovarian adenocarcinoma (OVCAR) cell line	Cervical and ovarian cancer	In cancer cells, epidermal growth factor siRNA could be delivered with efficiency.	[110]
11.	PEGylated and folate-targeted chitosan polymeric nanoparticles (FPNs)	Octadecyl quaternized carboxymethyl chitosan (OQC)	SGC-7901cells	Gastric carcinoma	The outcomes demonstrated that drug-resistant SGC-7901 cells could be reversed by folate-targeted chitosan polymeric nanoparticles (FPNs).	[111]
12.	chitosan	Interleukin-12	BALB/c mice, WEHI-164 tumor cells	Fibrosarcoma	In a mouse model of fibrosarcoma, IL-12 gene therapy with chitosan nanoparticles had therapeutic benefits on the regression of tumor masses.	[112]
13.	Hydroxyapatite coated with chitosan nanoparticles	Curcumin	U87MG cell line	Brain carcinoma	HA and chitosan have helped in the targeted delivery of curcumin, an anticancer agent.	[113]
14.	Chitosan coated with alginate	Cisplatin	Swiss albino mice.	Cervical cancer	Mucoadhesive spray-dried microparticles may offer a beneficial method for targeted delivery of anticancer treatment via the vaginal route for cervical cancer with increased effectiveness.	[114]
15.	Glycol chitosan	Small gold nanoparticles (sGNPs)	CT26 cancer cells and Balb/C mice	Colorectal carcinoma	Immunogenic and hyperthermal damage was observed in tumor cells resulting in cell death and prevention of cancer.	[115]
16.	Chitosan	5-Aminolevulinic acid (5-ALA) and photothermal reagent (IR780)	CT-26 cells	Colon cancer	Chitosan has the potential to manage colon cancer via oral administration.	[116]
17.	Fluorinated-chitosan	*meso*-tetra(4-carboxyphenyl)porphine-conjugated catalase (CAT-TCPP)	MB49 cells	Bladder cancer	Systematic toxicity helped in the treatment of bladder cancer	[117]
18.	Chitosan	Poly(γ-glutamic acid)	4T1 (orthotopic breast tumor mouse model)	Breast tumor	Chitosan nanoparticles conjugated with poly(γ-glutamic acid) could potentiate radiotherapy and act as an adjuvant in anticancer interventions.	[118]
19.	Pluronic grafted chitosan	Anti-HER2 monoclonal antibody	MCF-7 (human breast cancer cells) and Vero (kidney cell line of African green monkey)	Breast cancer	AntiHER2 conjugated with copolymer chitosan, and DOX can develop as a potential drug carrier for anticancer agents.	[119]
20.	Chitosan	Gold nanorods and DOX	MCF-7 cells, lung cancer A549 cells, Human cervical cancer HeLa cells and fibroblast L929 cells	Lung and cervical cancer	The cytotoxicity was observed against the tumor cells based on a combination of photothermal and chemical therapeutic activity of chitosan, DOX and gold nanorods.	[120]
21.	Carboxymethyl chitosan (CMC) and labelled fluorescein isothiocyanate (FITC)-chitosan hydrochloride (CHC) (FITC-CHC)-CMC	Anti-β-catenin siRNA	HT-29 cells	Colon cancer	The colon cancer cells’ formation of β-catenin protein was decreased to roughly 40.10% after 48 h of anti-β-catenin siRNA transfection, demonstrating a successful reduction in protein which encourages colon cancer proliferation. The findings showed that the siRNA-(FITC-CHC)-CMC delivery system has significant potential for RNAi therapeutical uses in cancer cells.	[121]
22.	Mucoadhesive chitosan	Oxaliplatin (OXPt)	SCC-9 (human tongue cancer cell line)	Oral tumors	The cells entering into apoptosis were increased by the usage of chitosan and resulting in treating oral tumors.	[122]
23.	Chitosan	tumor -targeting adenoviral (Ad), folic acid (FA) and poly(ethylene glycol) (PEG)	Folate receptor-positive human epithelial carcinoma cells from the oral cavity (KB), glioma cells (U343), human embryonic kidney cells (HEK293), and murine macrophage cells (RAW264.7)	Metastatic tumor treatment	Ad/chitosan-PEG-FA nanocomplexes dramatically reduced the inflammatory cytokine, IL-6, production from macrophages, suggesting a potential for systemic delivery. These findings unequivocally show that cancer cell-targeted viral transduction by Ad/chitosan-PEG-FA nanocomplexes could successfully treat metastatic tumors while minimizing immune response to Ad.	[123]
24.	Chitosan	Hydrogel microparticles	*VX2 carcinoma* model	Liver tumor	While (chitosan hydrogel) CHI embolization did not significantly impair liver function, it did decrease tumor development.	[124]
25.	O-carboxymethyl chitosan (O-CMC)	Metformin	MiaPaCa-2 (Pancreatic cancer cells)	Pancreatic cancer	Research revealed that such a unique strategy would overcome metformin’s present limitations in its therapeutic use against pancreatic cancer.	[125]
26.	Chitosan	MnFe_2_O_4_	MDA-MB 231 cancer cells	Breast cancer	The biocompatibility of chitosan-MnFe_2_O_4_ nanoparticles was extremely elevated, and thermal ability is an effectual agent for cancer treatment.	[126]
27.	Chitosan-pectinate	Curcumin	Pectinase (*Aspergillus niger*) and Mucin type III (Porcine stomach)	Colon cancer	The data strongly suggest that the system may be used as a mucoadhesive curcumin delivery method that is colon-targeted for the potential treatment of colon cancer.	[127]
28.	Chitosan	Copper oxide	A549 cancer cells	Lung cancer	The CS-CuO nanocomposite’s anti-proliferative effectiveness was assessed in the human lung cancer cell line A549. Against A549 cancer cells, the synthesized CS-CuO nanocomposite showed concentration-dependent anti-proliferative action.	[128]
29.	chitosan-g-methoxy poly(ethylene glycol) (ChitoPEG) copolymer	Thiodipropionic acid (TDPA) and phenyl boronic acid pinacol ester (PBAP)	CT26 mouse colorectal carcinoma cells	Colon cancer	ChitoPEG-PBAP nanophotosensitizer is a potential photodynamic candidate for cancer treatment.	[129]

## Data Availability

This is a review and the majority of the article’s references are cited appropriately in the manuscript.

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
