# Peer review of "Chitosan Nanoparticles-Based Cancer Drug Delivery: Application and Challenges"

_marinedrugs, 2023, doi:10.3390/md21040211_

Round 1

Reviewer 1 Report

This article gives an overview of the use of chitin and chitosan-based drug carriers in the treatment of cancer (and other conditions). After a brief introduction to the structure of chitin and a description of extraction methods used in its purification, the literature on chitin and chitosan based drug delivery formulations is summarised. This includes a variety of routes of administration in which the use of chitin/chitosan nanoparticles acts to increase the bioavailability of the active ingredient in the specific location required.

The substantial second part of the review concerns chitin and chitosan complexes in the treatment of cancer, providing a summary table of preclinical studies in cell and animal models. On the whole the review provides a useful guide to the literature of this field, which has expanded sharply over the past two decades.

There are some problems to deal with before publication.

1.      The structure of chitin is not well described in the introduction. No details are given for the monosaccharide composition or glycosidic linkages of the polysaccharide, and the phrase “the side chain’s backbone arrangement’ on line 59 is not easy to understand, as chitin does not have side chains. Perhaps “the arrangement of substituents along the backbone” is meant? The following paper has material on chitin structure:

Hou Jiaxin, Aydemir Berk Emre and Dumanli Ahu Gümrah 2021Understanding the structural diversity of chitins as a versatile biomaterial Phil. Trans. R. Soc. A.3792020033120200331 http://doi.org/10.1098/rsta.2020.0331

2.      The standard of English is on average not high in this paper, though it is variable; some parts can be read with ease and others are difficult. The authors need to find someone who can write English well and ask them to correct the text throughout.

3.      The references should be checked carefully; for example on line 194 the “study by Jana et al. [42]” is not correct as ref. 42 is Machalowski et al.; Jana is nowhere in the reference list.

4.      In Table 1, why is chemotherapy not included?

5.      Table 2 is a good way to present this material. Are there, in addition to the pre-clinical studies mentioned, any clinical trial results in humans? Are any chitin-based carrier formulations in clinical use? If any could be found they would make an interesting addition.  

Author Response

Reviewer 1:

This article gives an overview of the use of chitin and chitosan-based drug carriers in the treatment of cancer (and other conditions). After a brief introduction to the structure of chitin and a description of extraction methods used in its purification, the literature on chitin and chitosan based drug delivery formulations is summarised. This includes a variety of routes of administration in which the use of chitin/chitosan nanoparticles acts to increase the bioavailability of the active ingredient in the specific location required.

The substantial second part of the review concerns chitin and chitosan complexes in the treatment of cancer, providing a summary table of preclinical studies in cell and animal models. On the whole the review provides a useful guide to the literature of this field, which has expanded sharply over the past two decades.

There are some problems to deal with before publication.

1. The structure of chitin is not well described in the introduction. No details are given for the monosaccharide composition or glycosidic linkages of the polysaccharide, and the phrase “the side chain’s backbone arrangement’ on line 59 is not easy to understand, as chitin does not have side chains. Perhaps “the arrangement of substituents along the backbone” is meant? The following paper has material on chitin structure:

Hou Jiaxin, Aydemir Berk Emre and Dumanli Ahu Gümrah 2021Understanding the structural diversity of chitins as a versatile biomaterial Phil. Trans. R. Soc. A.3792020033120200331 http://doi.org/10.1098/rsta.2020.0331

Response: We thank the honourable reviewer for the encouraging comments and glad you found our manuscript informative. As suggested, we have now expanded the literature on the structure of chitin, and we have also included some of the points regarding the crystalline structure of chitin in the lines 68-79. Also, we really appreciate you for suggesting the paper for finding the information on the structure of chitin.

2. The standard of English is on average not high in this paper, though it is variable; some parts can be read with ease and others are difficult. The authors need to find someone who can write English well and ask them to correct the text throughout.\

Response: We agree with your suggestion and hence we have enhanced the quality of English in the manuscript. Therefore, the revised version of the manuscript has been submitted after thorough grammatical revisions and language improvements.

3. The references should be checked carefully; for example on line 194 the “study by Jana et al. [42]” is not correct as ref. 42 is Machalowski et al.; Jana is nowhere in the reference list.

Response: Thank you so much for correcting our mistake. We have revised the in-text citation according to the associated reference, line 217.

4. In Table 1, why is chemotherapy not included?

Response: Chemotherapy is indeed an important therapeutic approach for cancer. In the revised version of the manuscript, table 1 has been edited, and a section (no. 6) for “Chemotherapy” has been added with reference no. 85.

5. Table 2 is a good way to present this material. Are there, in addition to the pre-clinical studies mentioned, any clinical trial results in humans? Are any chitin-based carrier formulations in clinical use? If any could be found they would make an interesting addition.

Response: The addition of clinical trials would have embraced the manuscript. However, there are so far no studies on humans with cancer. Also, there are some challenges and development of toxicity for using chitosan on human beings. In line with your suggestion, we have added a new section (6.                   Challenges and future perspectives) elucidating the challenges and future perspectives of chitosan for drug delivery; lines 475-493.

Reviewer 2 Report

The manuscript is a review of chitosan nanoparticles especially for the cancer drug delivery. I think it is well organized and would like to recommend publication after revision.

1.  Two review articles can be added to “1. Introduction section” or “5. Chitin and chitosan for drug delivery and cancer treatment section” with some comments.

   Cancer Cell Int 2021 21:318https://doi.org/10.1186/s12935-021-02025-4

   Pharmaceutics 2017, 9, 53. doi:10.3390/pharmaceutics9040053.

2.  The following article can be added to “2.1 Chemical Process” with some comments.

   Scientific Reports 2022, 12:3515.  https://doi.org/10.1038/s41598-022-07073-y.

 Minor mistakes:

On page 3, “acetic” should be “acidic”.

On page 3, “CH3COO” should be “CH3COOH”.

On page 4, “CaHSO3” should be “Ca(HSO3)2”.

On page 4, Delete one of “Na3PO4”, it is duplicated.

On pages 4 and 18, “microwave-assisted methods (MWI)” should be “microwave irradiation (MWI)” according to the reference #42.

Author Response

Reviewer 2:

The manuscript is a review of chitosan nanoparticles especially for the cancer drug delivery. I think it is well organized and would like to recommend publication after revision.

1. Two review articles can be added to “1. Introduction section” or “5. Chitin and chitosan for drug delivery and cancer treatment section” with some comments.

Cancer Cell Int 2021 21:318https://doi.org/10.1186/s12935-021-02025-4

Pharmaceutics 2017, 9, 53. doi:10.3390/pharmaceutics9040053.

Response: We are immensely thankful to the honorable reviewer for the inspiring comments. As suggested, we have now submitted the manuscript for your perusal after doing all the necessary revisions and adding the articles mentioned by you. This has certainly improved the quality of both introduction and drug delivery and cancer treatment sections.

2. The following article can be added to “2.1 Chemical Process” with some comments.

Scientific Reports 2022, 12:3515.  https://doi.org/10.1038/s41598-022-07073-y.

Response: We have now updated our manuscript with the material from the excellent article suggested by the reviewer. This has significantly improved the material in sub section 2.1 chemical process, in the updated manuscript.

3. Minor mistakes:

On page 3, “acetic” should be “acidic”.

On page 3, “CH3COO” should be “CH3COOH”.

On page 4, “CaHSO3” should be “Ca(HSO3)2”.

On page 4, Delete one of “Na3PO4”, it is duplicated.

On pages 4 and 18, “microwave-assisted methods (MWI)” should be “microwave irradiation (MWI)” according to the reference #42.

Response: In the revised manuscript, we have made all corrections as suggested by the reviewer. We are heartly grateful to the reviewer for helping us in improving the manuscript by valuable constructive criticisms. 

Reviewer 3 Report

The aim of the review by Sachdeva and co-authors that have been brought to my attention is to provide a comprehensive overview of the innovations and progress in understanding the potential role of chitin and chitosan biopolymers in drug delivery for cancer treatment. This topic is not new and in the last years some reviews on the application of chitin and chitosan nanomedicine in cancer treatment have been published in other journals published by MDPI too (such as 10.3390/molecules27020473 and doi.org/10.3390/md20070460). Beyond the lack of originality of this review, it seems that the authors have missed the article's main point. Out of the five sections of the article, only one focuses on the use of chitin and chitosan in nanomedicine (section 5). The topics presented in sections 2, 3 and 4 have already been treated (even more accurately) in other reviews and are only of marginal interest to this one. Moreover, no mentions were made of the problematics of industrial preparation, administration, and toxicity of chitosan nanomedicines. Finally, the strategies to overcome the challenges of the use of these biopolymers in cancer therapy are not reported.

After careful consideration, I think this manuscript does not meet the minimum requirements to be published on Marine Drugs in this form

Author Response

Reviewer 3:

The aim of the review by Sachdeva and co-authors that have been brought to my attention is to provide a comprehensive overview of the innovations and progress in understanding the potential role of chitin and chitosan biopolymers in drug delivery for cancer treatment. This topic is not new and in the last years some reviews on the application of chitin and chitosan nanomedicine in cancer treatment have been published in other journals published by MDPI too (such as 10.3390/molecules27020473 and doi.org/10.3390/md20070460). Beyond the lack of originality of this review, it seems that the authors have missed the article's main point. Out of the five sections of the article, only one focuses on the use of chitin and chitosan in nanomedicine (section 5). The topics presented in sections 2, 3 and 4 have already been treated (even more accurately) in other reviews and are only of marginal interest to this one. Moreover, no mentions were made of the problematics of industrial preparation, administration, and toxicity of chitosan nanomedicines. Finally, the strategies to overcome the challenges of the use of these biopolymers in cancer therapy are not reported.

After careful consideration, I think this manuscript does not meet the minimum requirements to be published on Marine Drugs in this form

Response: We are immensely thankful for your valuable comments and suggestions on our manuscript. We agree with you that sections 2, 3 and 4 have been treated in depth in several more articles. Therefore, we have tried to accumulate and elucidate different articles on the “Route of chitosan administration” and “Chemical process” briefly. Moreover, as suggested by you, we have now submitted the article with a new section on “Challenges and future perspective” regarding chitosan drug delivery.

We hope that these changes enhance the quality of our manuscript and have reached the mark for publication in this esteemed journal. Further, we are heartly grateful to the reviewer for helping us in improving the manuscript by valuable constructive criticisms.

Round 2

Reviewer 3 Report

Despite the efforts made by the authors, in my opinion, this manuscript has not been improved in terms of content. The structure of this review has remained unchanged, still missing the focus on the advantages and problematics of using chitin and chitosan in nanomedicine. After this consideration, I still think that this manuscript does not meet the minimum requirements to be published on Marine Drugs in this form.

Author Response

Response to Reviewer Comment

Comment: Despite the efforts made by the authors, in my opinion, this manuscript has not been improved in terms of content. The structure of this review has remained unchanged, still missing the focus on the advantages and problematics of using chitin and chitosan in nanomedicine. After this consideration, I still think that this manuscript does not meet the minimum requirements to be published on Marine Drugs in this form.

Response: We thank the reviewer for the comment and suggestions on our manuscript. Unlike the manuscript submitted previously, in this submission, the content of the manuscript has undergone several major changes. Our team has changed the text, content and figures in the manuscript for more satisfactory clarity and understandability to the readers. Furthermore, we have added two more sections in the manuscript that are discussing the advantages and problematics of using chitin and chitosan in nanomedicine. After the second round of modifications that our team has made, we are quite sure that the manuscript is satisfactory for publication in this esteemed journal.

Round 3

Reviewer 3 Report

I appreciate the efforts made by the authors to improve the manuscript, but despite these efforts, in my opinion, the manuscript has not been improved in terms of content. I cannot endorse its publication on Marine Drugs.